



# The effect of sediment thermal conductivity on vertical groundwater flux estimates

Eva Sebok[1], Sascha Müller[1]

[1]Department of Geosciences and Natural Resource Management, University of Copenhagen, 1350, Copenhagen, Øster
Voldgade 10, Denmark

*Correspondence to*: Eva Sebok (es@ign.ku.dk)

**Abstract.** Vertical sediment temperature profiles are frequently used to estimate vertical fluid fluxes. In these applications using heat as a tracer of groundwater flow, the thermal conductivity of saturated sediments ($k_e$) is often given as a standard literature value and assumed to have a homogeneous distribution in the vertical space. In this study vertical sediment temperature profiles were collected both in a high-flux stream and a low-flux lagoon environment in a sand-, and peat-covered area. $k_e$ was measured at the location of each temperature profile at several depths below the sediment-water interface up to 0.5 m with a measurement spacing of 0.1 m. In general $k_e$ values measured in this study ranged between 0.55 and 2.96 W m$^{-1}$ °C$^{-1}$ with an increase with depth from the sediment-water interface. The effect of using a vertically homogeneous or heterogeneous distribution of measured $k_e$ values on vertical flux estimates was studied with a steady-state HydroGeoSphere model. In the high-flux stream environment estimated fluxes varied between 0.03 and 0.71 m d$^{-1}$ and in the low-flux lagoon between 0.02 and 0.23 m d$^{-1}$. It was found, that using a vertically heterogeneous distribution of sediment thermal conductivity did not considerably change the fit between observed and simulated temperature data compared to a homogeneous distribution of $k_e$. However, depending on the choice of sediment thermal conductivities, flux estimates decreased by up to 64% or increased by up to 75% compared to using a standard $k_e$ sediment thermal conductivity for sand, frequently assumed by previous local studies. Hence, our study emphasizes the importance of using spatially distributed thermal properties in heat flux applications in order to obtain more precise flux estimates.

## 1 Introduction

A thorough knowledge of exchange fluxes between groundwater and surface water is crucial for sustainable and responsible water management as groundwater flow is a pathway of transport for nutrients and pollutants to receiving surface waters. Moreover, groundwater also helps maintaining surface water ecosystems by providing a thermally stable environment or moderating the effect of climate change (Brunke and Gonser, 1997; Dahm et al., 1998; Hayashi and Rosenberry, 2002; Briggs et al., 2013; Kurylyk et al., 2015). More and more studies thus focus on groundwater-surface water exchange, the qualitative mapping of the main areas of exchange, the direction of groundwater flow and the quantification of the exchange



fluxes using various methods including seepage meters, hydraulic gradients, differential gauging and mass balance approaches (Kalbus et al., 2006; Rosenberry and LaBaugh, 2008).

In the past 10-20 years heat as a tracer also emerged as a way to quantify groundwater-surface water exchange. The method is based on the differences between the diurnally and seasonally variable surface water temperature and the relatively stable
groundwater temperature (Constantz, 2008). Advantages of the thermal methods are that heat is a robust tracer that can be inexpensively monitored (Kalbus et al., 2006) and sediment thermal properties vary over a narrower range than e.g. corresponding hydraulic properties (Stonestrom and Constantz, 2003, Anibas et al., 2011). The temperature distribution at the bed of surface water bodies can be used for qualitative mapping of potential discharge sites (Conant, 2004; Sebok et al., 2013; Briggs et al., 2011). Assuming only vertical flow, exchange fluxes between groundwater and surface water can be
quantified by point-scale vertical temperature profiles from the sediment bed either by fitting a steady-state analytical solution to the observed data (Schmidt et al., 2007; Anibas et al., 2011; Jensen and Engesgaard, 2011) or by time series analysis of sediment temperature data (Hatch et al., 2006; Keery et al., 2007; McCallum et al., 2012). Using observed temperature time series numerical models have also been used to calculate the direction and magnitude of groundwater fluxes (Karan et al., 2014).

Using either the steady-state analytical solution, time series analysis or numerical modelling to estimate vertical fluid flux, the thermal properties of sediments are most frequently assigned based on literature data (Schmidt et al., 2006; Hatch et al., 2006; Anibas et al., 2009; Jensen and Engesgaard, 2011; Anibas et al., 2011; Meinikmann et al., 2013). Thermal properties are rarely measured in the field and, due to their narrow range in values they are not expected to considerably influence flux estimates. However, Constantz et al. (2002) found that uncertainty in sediment thermal conductivity could lead to up to 50%
uncertainty in estimated channel percolation. Using time series analysis Shanafield et al. (2011) showed that uncertainty in sediment thermal properties could result in incorrect flux estimates especially in low-flux environments with upward flow. In such cases a decrease in temperature sensor spacing could reduce the uncertainty in thermal properties (Shanafield et al., 2011).

For some approaches sediment thermal conductivity ($k_e$) is not required to estimate groundwater flux, instead sediment
temperature time series are used to estimate sediment thermal diffusivity (McCallum et al., 2012; Luce et al., 2013). Thus, in case of unknown or poorly characterized thermal properties the solutions suggested by McCallum et al. (2012) and Luce et al. (2013) will most likely lead to more accurate flux estimates (Irvine et al., 2015). These solutions however require longer measurements of sediment temperature time series and are not suitable for quick mapping of larger areas often required in reconnaissance surveys.
Even though some authors reflect on the uncertainty of using standard values and a homogeneous distribution of $k_e$ (Shanafield et al., 2011), there are only very few studies where sediment thermal properties are directly measured in the field (Schmidt et al., 2007; Menichino and Hester, 2014; Halloran et al., 2017; Irvine et al., 2017) and even fewer where the horizontal heterogeneity of these thermal properties over the field site is taken into account (Duque et al., 2016). There are, however, some attempts where the vertical heterogeneity of $k_e$ is taken into account. Recently, Kurylyk et al. (2017)



presented a tool where the thermal conductivity of different material layers was incorporated in the solution when calculating vertical fluxes thus leading to more accurate vertical groundwater flux estimates. Yet, the majority of studies use uniform $k_e$ values obtained from literature.

Selecting an appropriate value of $k_e$ can be crucial in environments with low groundwater fluxes where conduction is dominating convection. Duque et al. (2016) found that using standard literature values based on sediment properties instead of in-situ measurements of $k_e$ resulted in a mean flux overestimation by 2.33 cm d$^{-1}$. At their low flux study site this overestimation corresponded to a mean increase of 89% in flux values. Yet, similar effects are expected at sites with high groundwater fluxes where convection dominates conduction. In a modelling study set in a high-flux environment, Karan et al. (2014) found that sediment thermal conductivity and vertical anisotropy in hydraulic conductivity were the most sensitive parameters influencing flux estimates. These studies highlight the need for an appropriate selection of $k_e$ both in low and high flux environments as it significantly influences vertical groundwater flux estimates.

There are a few studies where sediment thermal properties used in groundwater flux calculations are based on field measurements (Schmidt et al., 2007; Menichino and Hester, 2014) or where even individual measurements are made at the sediment surface for each sediment temperature profile (Duque et al., 2016). However, there is no comprehensive field study where the natural vertical variability in sediment thermal properties is explored within the shallow sediments of streams and lakes where sediment temperature profile measurements are routinely carried out. Therefore, the aims of this study were to: (1) assess the natural variability in the vertical distribution of $k_e$ in areas with different sediment properties; (2) characterize the range of vertical groundwater flux estimates using several vertical distributions of in-situ $k_e$ values measured at various depths at individual sediment temperature profiles, and (3) assess the effect of vertical heterogeneity in $k_e$ at both low and high-flux field sites at two different depositional environments.

## 2 Field sites

Field measurements were conducted at two field sites, one with relatively low upward groundwater fluxes (Duque et al., 2016) at Ringkøbing fjord and a second with relatively high groundwater fluxes (Poulsen et al., 2015; Karan et al., 2017; Jensen and Engesgaard, 2011) in Holtum stream in Western Denmark (Fig. 1a). Ringkøbing Fjord is a coastal lagoon with a brackish water (5-15‰ salinity), connected to the North Sea through a sluice at the barrier islands in the west. The coastal lagoon has an area of 300 km² and an average water depth of 1.9 m (Ringkøbing Amt, 2004). The water depth at the eastern shoreline, where the field measurements were carried out, is approximately uniform of 0.5 m depth. Haider et al. (2014) simulated groundwater discharge at the eastern shore of the lagoon and using seepage meters Müller et al. (2018) measured temporally variable discharge fluxes in response to recharge dynamics and spatial variability governed by sediment structure. Both studies found that the position of the saltwater-freshwater interface (Mulligan and Charette, 2006) had an effect on the groundwater fluxes. At the study site the sediment-water interface is characterized by organic sediments in the near-shore region, while further offshore medium-grained sand dominates. The shallow geology of the area is characterized by



Pleistocene fluvio-glacial sandy deposits intertwined by low permeable layers (Duque et al., 2016). In order to account for the differences between the organic deposits close to the shore and the sandy sediments further offshore, field measurements were carried out at an area covered by peat close to shore and in two areas in the sandy deposits further offshore between 10-11 June 2014.

The high-flux field site was located at the lowland, gaining Holtum stream, a headwater catchment of the Skjern river. The stream at the study site has a catchment area of 70.4 km² which is dominated by glacial sandy and silty deposits from the Weichselian glacial period (Houmark-Nielsen, 1989). The average annual stream discharge two km downstream of the study site was 1.2 m³ s⁻¹ for the period of 1994-2012 (Poulsen et al., 2015). At the study site the stream has a soft sandy streambed with mobile sediments (Sebok et al., 2015) consisting mainly of medium and coarse-grained sand and occasional organic

material (Sebok et al., 2014). Previous studies reported groundwater fluxes between 0.06 and 1.3 m d⁻¹ along several stream segments (Karan et al. 2017, Poulsen et al. 2015). Field measurements at the stream site were carried out on 11-12 August 2014 in a straight stream section of 3 m length and 3.5 m width in several transects across the stream (Fig. 1b). The stream water depth at the measurement locations varied between 0.8-1.15 m.

### 3 Methods

**3.1 Field measurements**

Sediment temperatures were recorded at several depths (0, 5, 10, 15, 20, 35, 50 cm depth) below the sediment surface using PT100 resistance thermometers installed with the direct push technique. After a stabilization time of 30 seconds, the temperatures were recorded with an accuracy of 0.2 °C. Vertical sediment temperature profiles were measured at 12 sites in the stream on 11-12 August 2014 (Fig. 1b) and at 19 sites in the lagoon on 10-11 June 2014 (Fig. 1c). Out of 19 sites in the

lagoon, 5 were located in the peat-, and 14 in the sand-covered area.

Immediately after the collection of vertical sediment temperature profiles, sediment thermal conductivity was measured on site by the KD2 probe using the SH-1 sensor (Decagon Devices, Pullman, WA, USA) on sediment cores taken from the same location. The device measures thermal conductivity with a ±10% accuracy in the range of 0.2 to 2 W m⁻¹ °C⁻¹. To obtain these measurements first a plastic PVC pipe of 5 cm outer diameter, open at both ends, was inserted in the streambed

as deep as possible, but always deeper than 50 cm. Then sediment cores trapped in the PVC pipes were collected by creating vacuum in the pipes by the aid of a vacuum pump and carefully removing them from the streambed. A plastic cap was inserted at the bottom of the sediment cores thereby trapping the sediments and the surface water column above the sediments in the PVC pipes providing for fully saturated conditions during the measurement of thermal conductivity. The top of the PVC pipes were gradually cut at several heights, thus thermal conductivity could be measured at specific depth levels

in the saturated sediment column by inserting the sensor in the exposed upper sediment layers. A similar setup, with a larger pipe diameter, was also used by Smits et al. (2016) under laboratory conditions to measure the thermal conductivity of soils. Before the field measurements, laboratory tests were conducted to establish the influence of the pipe diameter on the





measurements. It was found that this pipe diameter does not have any influence on measurements if the needles of the KD2 probe are inserted vertically in the trapped sediment column.

The thermal conductivity of saturated sediments was measured with a 10 cm vertical interval up to 50 cm depth below the sediment bed. This vertical interval and deployment depth are in the ranges of widely used vertical spacing of sensors

measuring temperature in the sediments (Schmidt et al., 2006; Hatch et al., 2006; McCallum et al., 2012). Due to operational challenges, it was not always possible to remove the sediment cores or the full length of the trapped sediments. Sometimes the sediment core became unsaturated and the corresponding vertical temperature profiles were omitted from the analysis. Thus, sediment temperature data and thermal conductivity in a vertical profile were analyzed at 7 sites in the stream, and in 5 peat-covered as well as 9 sand-covered locations in the lagoon (Fig. 1b, c). During each measurement, the KD2 probe also

calculated the measurement error. Measurements with an error larger than 0.05 W m$^{-1}$ °C$^{-1}$ were removed from the analysis.

**3.2 Data analysis and numerical modelling**

The similarity of saturated thermal conductivity measured at different sites, sediments and depths was assessed by the aid of the nonparametric Kruskal-Wallis test with a significance level of $p < 0.05$.

As the field measurements were carried out in June and August, it was assumed that a steady-state solution is applicable to

estimate vertical groundwater fluxes. A steady-state heat transport model was set up in HydroGeoSphere with a model domain of 1 m in each direction and a discretization of 2.5 cm in the vertical direction. A vertical hydraulic conductivity of 1 m d$^{-1}$ and a porosity of 0.3 was assigned to the model domain. The measured temperature at the sediment surface (0 cm depth) was used as a boundary condition at the top of the model domain, while a temperature of 11.5 °C was implemented for groundwater at the lagoon and a 9.45 °C at the stream site with the exception of profile H4 where a groundwater

temperature of 8.2 °C was applied. Vertical groundwater fluxes were obtained with PEST by minimizing the difference between the observed vertical sediment temperatures and sediment temperatures simulated by the model.

The role of sediment thermal conductivity on estimated fluxes was assessed by assigning various thermal conductivity values to the model layers. For each measurement location, vertical groundwater fluxes were estimated using five different distributions of $k_e$. In the first four cases $k_e$ was assumed to be homogeneous in the model domain, while the last case

represents a vertically heterogeneous, layered distribution of $k_e$. In the first homogeneous case a $k_e$ value of 1.84 W m$^{-1}$ °C$^{-1}$ frequently used by local studies (Jensen and Engesgaard, 2011; Duque et al., 2016, Poulsen et al., 2015), corresponding to saturated sand (Lapham, 1989; Stonestrom and Constantz, 2003), was applied (Case 1). In the subsequent homogeneous cases fluxes were estimated using the average (Case 2), minimum (Case 3), and maximum (Case 4) of the measured $k_e$ values within the individual profiles. This was done to to assess the range of groundwater fluxes that can be obtained using

in-situ measured sediment thermal conductivity. For the heterogeneous case a vertically heterogeneous distribution of $k_e$ was assigned to the model using the $k_e$ values measured from the top of the sediment layer with 10 cm intervals (Case 5). The $k_e$ value measured at the top of each depth level was assigned to the 10 cm layer below the measurement and the $k_e$ value measured at the deepest sediment level was assigned to the sediments up to bottom of the model domain at 1 m depth.





The models with the different $k_e$ distributions (Case 1-5) were run to steady state and the influence of $k_e$ on vertical flux estimates was evaluated by comparing the range of fluxes obtained for each individual profile. The effect of using a homogeneous or a heterogeneous vertical distribution of sediment thermal conductivity was assessed by comparing the Root Mean Square Error (RMSE) of observed and simulated sediment temperatures.

## 4 Results

### 4.1 Natural variability in sediment thermal conductivity

The measured thermal conductivity of saturated sediments across all profiles and materials ranged between 0.55 and 2.96 W $m^{-1}$ $°C^{-1}$ (Table 1). Maximum values measured at the stream and the two lagoon sites were similar, ranging from 2.72 to 2.96 W $m^{-1}$ $°C^{-1}$, while minimum values showed a larger spread ranging from 0.55 W $m^{-1}$ $°C^{-1}$ at the stream site, 0.65 W $m^{-1}$ $°C^{-1}$

in the peat of the lagoon, and up to 1.20 W $m^{-1}$ $°C^{-1}$ in the sand at the lagoon (Table 1). Pooling the $k_e$ values measured in all profiles at all depths, the Kruskal-Wallis test did not indicate a statistically significant difference between the pooled thermal conductivity values measured at the three different sites: sand in the stream, peat in the lagoon and sand in the lagoon.

The distribution of thermal conductivity showed a general increasing tendency with depth from the sediment-water interface (SWI) (Fig. 2), with the largest variability close to the SWI. At the lagoon sites, an initial increase in $k_e$ is followed by

approximately stable $k_e$ values at 0.1m and 0.3 m depth at the peat and sand locations, respectively. In contrast, at the stream site $k_e$ increased steadily with depth up to the measured depth of 0.5 m below the SWI (Fig. 2). Pooling data from all sites and all profiles together according to their measurement depth, the Kruskal-Wallis test showed a statistically significant difference between $k_e$ values at the SWI and measurements at 0.1m depth as well as the SWI and >0.3 m depth. There were no statistically significant differences between measurement depth of 0.2 and deeper observations (0.3-0.5 m). These results

are in accordance with Fig. 2 showing an increase until a specific depth after which $k_e$ remains approximately stable.

Comparing thermal conductivity values measured at different sites at specific depths below the SWI, the Kruskal-Wallis test showed a statistically significant difference between the $k_e$ values measured in the lagoon peat at the SWI and 0.3 m below and in the lagoon sand at the SWI and 0.3 and 0.4 m below. At the stream site the only statistically significant difference was indicated between depths of 0.1 and 0.5 m below the SWI, but due to the low sample count (n=2) at 0.5 m depth below the

streambed this result is not considered to be representative.

### 4.2 Vertical groundwater flux estimates

The steady state numerical model performed best at the high-flux stream site (Fig. 3). Here the best fit between the measured and simulated data was achieved at profile H5 with an RMSE of 0.02 °C, while the worst fit occurred at profile H4 with an RMSE of 0.32 °C. In the low-flux lagoon the best fit was achieved at the sand-covered area at profile S4 with an RMSE of

0.14 °C and in the peat-covered area at P4 with 0.28 °C. The worst fits were achieved at profile P1 in the peat-covered and at S7 in the sand-covered area with RMSEs of 0.74 and 0.75 °C respectively. Using a homogeneous (Case 1-4) or



heterogeneous (Case 5) vertical distribution of sediment thermal conductivity did not influence the fit between the measured and simulated temperature distributions considerably (Fig. 3).

Considering flux estimates with all five distributions of sediment thermal conductivity, vertical groundwater fluxes in the high-flux stream environment were between 0.03 and 0.71 m d$^{-1}$ (Table 2). The lowest fluxes were estimated at H4, where,
as opposed to the other profiles, a groundwater temperature of 8.2 °C had to be assigned in order to achieve a reasonable fit between the observed and simulated sediment temperatures. At this profile the variability of flux estimated with different distributions of $k_e$ is also the lowest at the stream site (Fig. 4). Estimated groundwater fluxes in the lagoon, in the low flux environment, ranged between 0.02 and 0.23 m d$^{-1}$ (Table 2) with generally higher fluxes and higher spatial variability of fluxes at the peat-covered area (Fig. 4).

There was a clear difference between the spatial variability of estimated fluxes in the low-, and high-flux environment with the high-flux stream environment generally displays a larger spatial variability in fluxes among measured profiles than at the low-flux lagoon (Fig. 4) and also a larger variability depending on the distribution of saturated sediment thermal conductivity in the model. The 95% confidence bounds on the flux estimates were tightest in the sand-covered lagoon areas. This cannot exclusively be related to the low-flux environment as fluxes estimated in the peat-covered area with the
minimum measured $k_e$ values are comparable in magnitude, but still have wider confidence bounds. Similarly, even though the flux estimates were the highest at the stream site, the confidence bounds on these flux estimates were comparable with the peat-covered area in the low-flux lagoon.

Flux estimates showed a considerable variability as a function of $k_e$ value and vertical distribution of the sediment thermal conductivity (Fig. 4). In Case 1 ($k_e = 1.84$ W m$^{-1}$ °C$^{-1}$), estimated fluxes at the high-flux stream site ranged between 0.05 and
0.44 m d$^{-1}$, while in the low-flux environment in the lagoon fluxes between 0.04-0.17 m d$^{-1}$ were found (Table 2 and Fig. 4). Generally, assigning the lowest thermal conductivity measured in the individual profiles for the entire length of the model domain (Case 3) resulted in the lowest flux estimates, with fluxes between 0.03-0.35 m d$^{-1}$ in the stream and 0.02-0.11 m d$^{-1}$ at the lagoon sites (Fig. 4). Compared to Case 1, this corresponded to a mean decrease of 26% and 44% in calculated fluxes in the lagoon and the stream, respectively (Table 3). Case 3 also lead to the smallest spatial variability of flux estimates
within the studied area and the smallest confidence bounds of the individual sites. Assigning the maximum measured $k_e$ (Case 4) resulted in the highest flux estimates, with fluxes of 0.04-0.71 m d$^{-1}$ in the stream and 0.06-0.23 m d$^{-1}$ in the lagoon translating into a mean increase of 41% and 36% compared to Case 1 in the lagoon and stream, respectively (Table 3). Yet, Case 4 also gave the highest spatial variability of estimated fluxes and largest confidence bounds at the individual measurement locations (Fig. 4). Assigning the average of measured $k_e$ (Case 2) for the respective profiles generally gave flux
estimates close to flux estimates of Case 1 with fluxes between 0.05-0.16 m d$^{-1}$ in the lagoon and 0.03-0.53 m d$^{-1}$ in the stream. A mean difference compared to Case 1 could only be observed in the lagoon sites, where estimated fluxes increased on average by 12% (Table 3).

Flux estimates obtained by assigning a vertically variable $k_e$ to the entire model domain (Case 5) gave different results in the low-flux lagoon compared to the high-flux stream environment. For the lagoon site all 5 profiles in the peat-covered and 9



profiles in the sand-covered area gave flux estimates close, yet slightly lower than using the maximum measured $k_e$ value (Case 4). This translated into flux estimates between 0.06-0.23 m d$^{-1}$ (Table 2) giving a mean increase of 28% compared to Case 1 (Table 3). In the high-flux stream environment, a vertically heterogeneous distribution of $k_e$ lead to an estimated flux range of 0.03-0.64 m d$^{-1}$ (Table 2) and a mean increase of 15% in fluxes (Table 3). As opposed to the lagoon, it did not result

in consistent changes in flux estimates. At profiles H2, H6 and H10 it approximately gave the same results as using the maximum $k_e$ measured in the profiles. In H1 it was closest to the estimates of using the maximum $k_e$, while at profiles H4 and H5 it agreed well with using the minimum measured $k_e$ values (Fig. 4). At the last remaining profile a vertically heterogeneous distribution of $k_e$ gave flux estimates closest to the measured average $k_e$ of the profile.

## 5 Discussion

### 5.1 Method assessment

The results of the present study are subject to several uncertainties both in the field measurements and in the numerical solution. In the lagoon samples plant roots occasionally occurred in the sediment, reducing thermal conductivities. Plant roots are an important source of organic matter (Angers and Caron, 1998) which in turn is known to decrease sediment thermal conductivity (Abu-Hamdeh and Reeder, 2000). If the presence of roots under the sediment layer was noticed, the

measurement was repeated by avoiding or removing the roots, hence a slight disturbance of the upper sediment layers may have occurred. By tilting the PVC pipes during their removal from the sediment bed, the topmost few centimeters of the trapped sediment column could also be occasionally disturbed. In-situ measurements of thermal conductivity could also be influenced by strong groundwater fluxes which cause changes in temperature conditions around the measurement device. However, as in this study the sediment cores were removed prior to the measurements, this potential uncertainty can be

excluded in this study.

Due to logistical reasons, measurements of $k_e$ were collected in the field with 10 cm measurement interval. These data were assigned to the vertically heterogeneous model with the assumption that the measured values are representative of the saturated sediment column up to 10 cm below the measurement with a homogeneous distribution of sediment thermal conductivity in that 10 cm sediment layer. The distribution of $k_e$ with depth shows that after an initial increase, $k_e$ values are

approximately stable at 0.1 m below the SWI in the peat-covered, and 0.3 m below the SWI at the sand-covered area of the lagoon, but changes considerably with depth at the stream site (Fig. 2). This suggests that the natural variability in sediment thermal conductivities in the vertical space may be different, likely even higher than presented in this study.

The vertical temperature distribution in the saturated sediments was simulated by a steady-state numerical model assuming vertical groundwater flow. At the stream site the vertical flow component is high enough to neglect the influence of the

horizontal flow component. However, at the low-flux lagoon site, the sediment temperature distribution could be influenced by a horizontal flow component. The model also assumed steady-state conditions which previously have been shown to be valid at the high-flux stream site where groundwater showed very damped seasonal temperature fluctuations (Poulsen et al.,





2015; Jensen and Engesgaard, 2011). Yet, in the low-flux lagoon environment, the diurnal temperature changes may influence the upper boundary condition of the sediment temperature profiles and groundwater temperature has a larger seasonal variability than at the high-flux stream site. Moreover, groundwater fluxes in coastal areas may also be diurnally variable due to the wave pumping effect (Rosenberry et al., 2013) and show variations on a larger temporal scale following

changes in the location of the freshwater-saltwater interface (Mulligan and Charette, 2006). The differences between the high-flux stream and low-flux lagoon sites are also reflected in the modelling results, with the high-flux stream site having a much better visual fit and lower RMSE closely approximating the accuracy of the temperature sensors as opposed to the low-flux lagoon site (Fig. 3).

Vertical groundwater flux estimates of this study are presented with their 95% confidence interval (Fig. 4). This confidence

limit, however, only encompasses uncertainties in the steady-state model, but does not incorporate the uncertainty of field measurements, where sediment temperature data was recorded with an accuracy of 0.2 °C and sediment thermal conductivity was measured with 10% accuracy. Thus, it is assumed that the 95% confidence interval on the flux estimates is even larger than presented in the study.

### 5.2 Natural variability in sediment thermal conductivity

Sediment thermal conductivities measured in this study ranged between 0.55-2.96 W m$^{-1}$ °C$^{-1}$ at the stream site and between 0.65-2.91 W m$^{-1}$ °C$^{-1}$ at the lagoon site (Table 1). The measured conductivity range corresponds to a range of organic sediments to sand (Lapham, 1989), whereas values between 0.8 and 2.5 W m$^{-1}$ °C$^{-1}$ are generally assumed for natural sediments (Hopmans et al., 2002; Stonestrom and Constantz, 2003). Measurements made in this study, however, also cover values larger than previously measured in field conditions or assumed in studies. An explanation for this could be that

measurements in this study were also made at other depths below the SWI, where thermal conductivity values show a generally increasing trend with depth, most likely reflecting a transition of finer to coarser sediments. Even though such higher values were not previously reported in field studies, similarly high values are frequently used in modelling studies (Schmidt et al., 2007; Karan et al., 2014). Previously, Duque at al. (2016) also measured thermal conductivities between 0.62-2.19 W m$^{-1}$ °C$^{-1}$ at the lagoon surface, while in our study values between 0.65 and 1.99 W m$^{-1}$ °C$^{-1}$ were found at 0 m

depth at the lagoon surface.

The vertical profiles of sediment thermal conductivity measured in the field at different sites and different sediments showed a horizontally and vertically heterogeneous distribution with increasing thermal conductivities with depth (Fig. 2). Thus, these findings contradict the common assumption of constant $k_e$ over the vertical sediment profiles when calculating vertical groundwater fluxes. Furthermore, the lower thermal conductivity in shallow depths suggests that the upper sediment layers,

close to the SWI are composed of generally finer sediments and/or contain more organic matter. This zone, also encompassing the root zone of aquatic vegetation, could be visually confirmed in the lagoon sediments, especially at the peat-covered area where plant roots were frequently visible in the sediment column.





The observed vertical distribution of finer upper sediment layers underlain by coarser materials also can be explained by general sedimentary processes where the fine material of sediment beds is easier to mobilize and redeposit than coarse grained sediments, thus overlaying coarse grained sediments observed at the lower part of sediment profiles. Moreover, in the peat-covered area of the lagoon the root zone of aquatic vegetation is located in the upper part of the sediment columns

(Duque et al., 2016). A similar vertical distribution of calibrated sediment thermal conductivity, with lower conductivity values in the upper and higher conductivity values in the lower layers, was also used by Naranjo et al. (2012) in a modelling study reporting values of 0.50-1.52 W m$^{-1}$ °C$^{-1}$ for a shallow and 0.86-2.68 W m$^{-1}$ °C$^{-1}$ for a deep streambed zone.

Sediment thermal conductivity not only increased with depth, but also reached a stable value at a specific depth in the lagoon sediments (Fig. 2), approximately at 0.1 m depth below the SWI at the peat-covered and 0.3 m depth below the SWI in the

sand-covered area. This distinction was confirmed by the Kruskal-Wallis test showing a statistically significant difference between $k_e$ measured at the SWI and the depths below 0.3 m below the SWI in the lagoon sediments. At the same time it must also be taken into account that due to the logistical difficulties, more measurements were available from the shallow depths (n= 18 at the SWI, while n= 9 at 0.5 m depth from SWI for all measurement profiles), thus the smaller sample size at greater depths may add bias to the results. At the high-flux stream environment the only statistically significant difference

between measurement depths was observed between the SWI and 0.5 m depth, most likely due to a gradual change in thermal conductivity with depth (Fig. 2). However, the results must be considered with caution as only two measurements were available at 0.5 m depth.

In the peat-covered area of the lagoon low $k_e$ values were expected due to the higher content of organic matter. Field observations however, do not agree with this assumption. Even though the largest portion of organic matter and roots were

observed in the peat-covered lagoon area, $k_e$ becomes already approximately stable at 0.1 m below the SWI (Fig. 2). This is considered a shallow depth as opposed to the stream sediments where even though no organic matter was visually detected, $k_e$ did not reach stable values in the measured 0.5 m long profiles (Fig. 2). Such contradiction may be explained by the difference in sediment structure and depositional environment at the field sites. At the stream site a previous study found a layered sediment structure with three sediment layers up to 0.5 m below the SWI which was rearranged between

measurement periods several months apart (Sebok et al., 2014). That study concluded that in the dynamic environment of a stream, sediments can be eroded up to a considerable depth below the SWI during high-discharge events. This may explain the greater vertical variability in $k_e$ in the stream environment as opposed to the lagoon, where sediments are not redistributed up to such a great depth and frequency even though erosional processes may also to influence $k_e$ at the lagoon site. For example, wave action may disturb sediments in the upper part of the lagoon bed. Such disturbances are mainly

expected in the sand-covered area, while vegetation reduces the effect of wave action in the peat-covered area of the near shore region (Fig. 1c). This difference in the depositional environments agrees well with the vertical distribution of $k_e$, where the stream, the sand-covered lagoon site and the peat-covered lagoon site are decreasingly dynamic. Accordingly, the stream site did not reach an approximately stable $k_e$ value in 0.5 m and in the peat-covered area $k_e$ becomes quasi-stable at





approximately 0.1m below the SWI. Based on this, it is also assumed that the zone of stable sediment thermal conductivity indicates a depth below the SWI where sediments are not eroded and redistributed by dynamic surface processes.

The results of this study also show that the sediment composition under the lagoon is not as diverse as expected. In greater depths below the SWI in the peat covered area, the measured $k_e$ values correspond to sand (Lapham, 1989) and agree with
the values measured in the sand-covered area in similar depths. This suggests that even though the top of the sediment profiles is dominated by peat and organic sediments, the lower part of the profile is most likely composed of sand.

### 5.3 Effect of sediment thermal conductivity on flux estimates

Upward groundwater flux estimates were between 0.03-0.71 m d$^{-1}$ at the stream site and 0.02-0.23 m d$^{-1}$ at the lagoon sites (Table 2, Fig. 4). The range of flux values agree well with previously published data from the stream site (Poulsen et al.,
2013; Karan et al., 2017), yet fluxes are slightly lower than reported by those studies. Using a range of different thermal conductivity values measured at the lagoon bed surface, Duque et al. (2016) reported fluxes up to 0.1 m d$^{-1}$ in the lagoon which are lower than flux values found in the present study. Reasons are to be found in the specific groundwater discharge pattern of the lagoon which is also closely related to changes in recharge conditions (Müller et al., 2018) and saline wedge location (Mulligan and Charette, 2006). Additionally, the manual calibration approach for the analytical solution chosen by
Duque et al. (2016) may also cause some differences to the automated calibration by PEST applied in the present study as with manual calibration special weight can be given to specific parts of the temperature profile, while with PEST all observations were weighted equally in this study.

This study also found that there is a difference in the magnitude of upward groundwater fluxes between the peat-covered and sand-covered area of the lagoon. Except for using the minimum measured thermal conductivity at the individual profiles,
upward groundwater fluxes are generally higher in the peat-covered area (Fig. 4), contrary to the previous expectations of having higher fluxes in sand. Yet, this study showed that the thermal conductivity of sediment columns in the peat-covered area is very similar to sand sediments (Fig. 2) making it likely that even in the peat-covered area the majority of sediments is composed of sand. Both the peat-covered and sand-covered area are dominated by sandy sediments with higher upward fluxes in the near-shore area. This agrees with common perception of exponentially decreasing groundwater fluxes in the
offshore direction under homogeneous sediment conditions (McBride and Pfannkuch, 1975).

The average of sediment thermal conductivity values measured in this study in different materials compares well with the standard literature values for sand (Table 1). Thus, using the average $k_e$ values measured in the individual profiles (Case 2) and the average literature value for sand (Case 1) gives similar flux estimates (Fig. 4). Using a vertically heterogeneous distribution of $k_e$ values in the model domain (Case 5) gave flux estimates close to using the maximum of measured $k_e$ values
(Case 4), especially in the lagoon (Fig. 4). A reason for this could be that $k_e$ reached a relatively stable value in a shallow depth from the SWI (Fig. 2), therefore the average $k_e$ of profiles is biased towards the higher values observed at the lower part of profiles. Similarly, this bias could explain the inconsistency in different flux estimates in the stream environment, where $k_e$ values increase with depth from the SWI, but do not reach a stable value.





Based on the results of this study, the choice of $k_e$ and its distribution did not improve the fit between observed and simulated temperature profiles substantially (Fig. 3) even though there is a large difference between flux estimates using different values and vertical distributions of $k_e$ (Fig. 4). It is assumed that other factors such the assumption of steady state conditions as well as only a vertical flux component has more effect on the fit than the choice of $k_e$ (Karan et al., 2013; Jensen and Engesgaard, 2011). Kurylyk et al. (2017) found distinct, visible differences in the shape of vertical sediment temperature profiles when incorporating sediment layers with different thermal conductivities in a model. However, Kurylyk et al. (2017) used very sharp boundaries within different material properties, while in this study due to the closely spaced vertical sampling, the transition between layers of different thermal properties was more gradual, possibly due to the narrow spacing of layers. Even though, using field measurements at several sites, this study confirmed a large vertical heterogeneity in sediment thermal conductivity, the vertical measurement interval of 10 cm used in this study is most likely more dense than necessary to capture the characteristic vertical heterogeneity in sediment layers. Based on the results of this study, it is however recommended to use representative $k_e$ values for each distinct sediment layer found at the field site.

Using various in-situ measured $k_e$ values gave a wide range of vertical flux estimates (Fig. 4) emphasizing the importance of using values representative for individual field sites to obtain correct flux estimates. The present dataset shows that using in-situ measured $k_e$ values, vertical groundwater fluxes could be up to 64% lower or 75% higher than flux estimates using standard $k_e$ values for sand (Table 3). Duque et al. (2016) also reported up to 89% increase in fluxes when using in-situ measured sediment thermal conductivities. Agreeing with conclusions of previous studies focusing on the sensitivity of flux estimates (Constantz et al., 2002; Kurylyk et al., 2017), the choice of a representative $k_e$ value can be crucial for flux estimates based on thermal gradients both in conduction and convection dominated environments.

## 6 Conclusions

This study investigated the natural vertical variability in sediment thermal conductivity measured in situ at a stream and a lagoon site within sandy and peat-covered sediments. Moreover, it analyzed the influence of the magnitude and vertical distribution of $k_e$ on vertical groundwater flux estimates both in a low-flux and a high-flux environment. Measured $k_e$ values ranged between 0.55 and 2.96 W m$^{-1}$ °C$^{-1}$ and showed a general increase with distance from the SWI until reaching an approximately stable value deeper below the SWI. Hence, this study shows both a horizontal and vertical spatial variability even over 0.5 m depth from the SWI. The depth of stable thermal conductivity values was related to the sedimentary environment, with the low-energy peat environment of the lagoon reaching a stable value 0.1 m below SWI, while in the dynamic stream environment no stable values were reached. $k_e$ influenced flux estimates significantly, by up to 75% compared to using widely applied standard values representative of sand. Vertical groundwater flux estimates ranged between 0.03 and 0.71 m d$^{-1}$ in the high-flux stream and 0.02 and 0.23 m d$^{-1}$ in the low-flux lagoon environment. The detected large vertical variability of $k_e$ values even over 0.5 m distance from the SWI and the large range of obtained vertical





flux estimates suggests that the selection of a representative sediment thermal conductivity value for each sediment layer is crucial for obtaining correct groundwater flux estimates.

### Acknowledgements

The study was supported by the Centre for Hydrology (HOBE) funded by the Villum Foundation.

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

| | Lagoon | | Stream |
|---|---|---|---|
| | Peat | Sand | Sand |
| Minimum | 0.65 | 1.20 | 0.55 |
| Maximum | 2.91 | 2.72 | 2.96 |
| Average | 2.07 | 2.16 | 1.86 |

**Table 1: Summary of measured thermal conductivity ($k_e$) in different sediment types at the two field sites. The values are given in W m$^{-1}$ °C$^{-1}$.**



| | Case 1 | | Case 2 | | Case 3 | | Case 4 | | Case 5 | |
|---|---|---|---|---|---|---|---|---|---|---|
| | Stream | Lagoon | Stream | Lagoon | Stream | Lagoon | Stream | Lagoon | Stream | Lagoon |
| Average | 0.27 | 0.09 | 0.29 | 0.10 | 0.16 | 0.06 | 0.40 | 0.13 | 0.29 | 0.12 |
| Minimum | 0.06 | 0.05 | 0.03 | 0.05 | 0.03 | 0.02 | 0.04 | 0.06 | 0.03 | 0.06 |
| Maximum | 0.44 | 0.17 | 0.53 | 0.16 | 0.35 | 0.11 | 0.71 | 0.23 | 0.64 | 0.23 |
| Standard deviation | 0.13 | 0.04 | 0.17 | 0.04 | 0.12 | 0.02 | 0.22 | 0.05 | 0.22 | 0.05 |

**Table 2: Summary of the average, minimum, maximum and standard deviation of flux estimates at the two field sites using different distributions of measured sediment thermal conductivity in the individual profiles: a homogeneous distribution of standard literature values (Case 1), the average (Case 2), minimum (Case 3) and maximum(Case 4) measured values of the individual profiles and a vertically heterogeneous distribution of measured data (Case 5). Upward groundwater flux values are given in m d$^{-1}$.**





|  | Case 2 | Case 3 | Case 4 | Case 5 |
|---|---|---|---|---|
| Lagoon_avg | -26 | +41 | +12 | +28 |
| Lagoon min | -63 | +24 | -10 | +16 |
| Lagoon max | +8 | +75 | +24 | +40 |
| Stream_avg | -44 | +36 | 0 | +15 |
| Stream_min | -64 | -35 | -39 | -52 |
| Stream_max | -9 | +61 | +27 | +57 |

**Table 3: Average, minimum and maximum percentage changes in estimated vertical groundwater fluxes compared the using a standard thermal conductivity value of 1.84 W m$^{-1}$ °C$^{-1}$ representative of sand and traditionally used in local studies.**



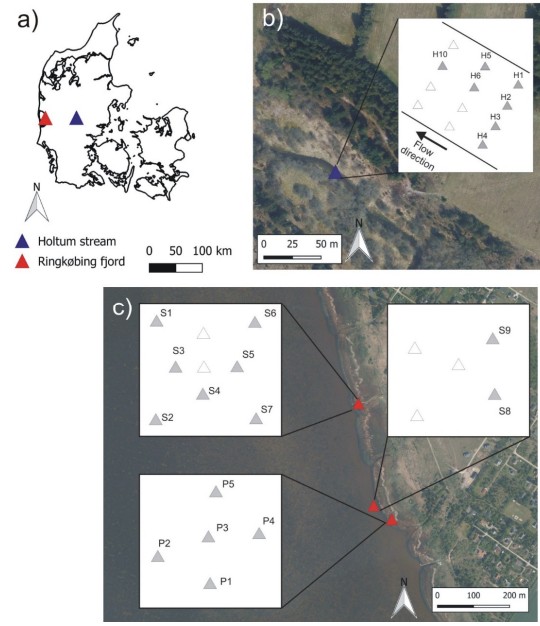

**Fig. 1:** Location of the field sites in Denmark (a) with the location of the profiles in Holtum stream (b) and Ringkøbing fjord (c).
On panels b) and c) the triangles mark the locations where vertical sediment temperature profiles were measured, while the grey
triangles indicate the profiles where sediment thermal conductivity was measured as well. Data from the locations marked with a
5   grey triangles are discussed in this study.

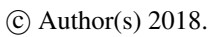



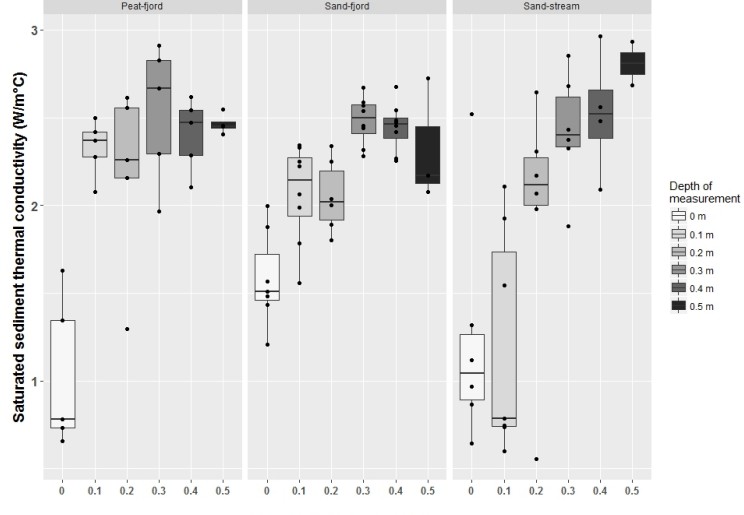

Fig. 2: Box plot of measured sediment thermal conductivity values at each site over depth.

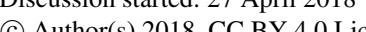



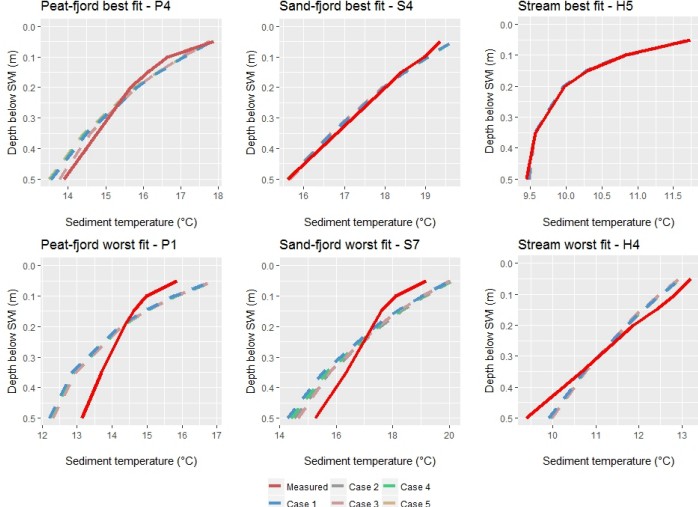

Fig. 3: Best and worst fit between measured and simulated temperature values at the field sites.



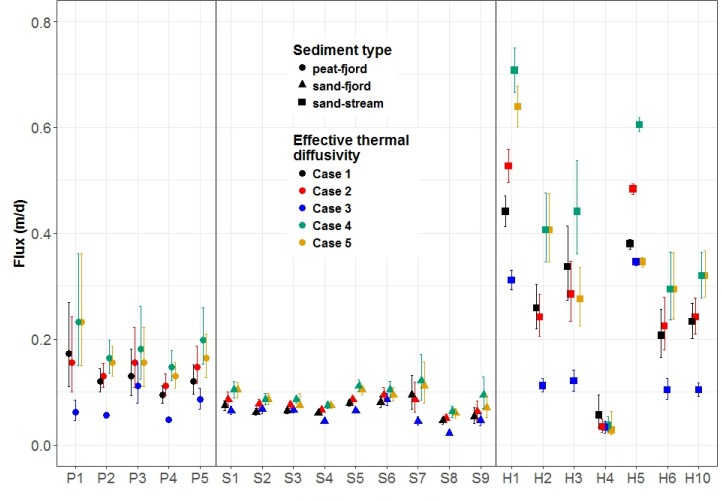

**Fig. 4: Vertical groundwater fluxes estimated at the test sites by assuming various distributions of sediment thermal conductivity (Case 1-5). Sites with identification of P refer to the peat-covered area in the lagoon, S to the sand-covered area in the lagoon and H to the stream site (Fig. 1b,c).**