# Peer review of "The effect of sediment thermal conductivity on vertical groundwater flux estimates"

_Hydrology and Earth System Sciences, 2018_

## Referee Comment (RC1) · Anonymous Referee #1 · 29 May 2018

General comments: The paper presents an evaluation of the influence of vertical thermal conductivity variability on the estimates of vertical GW-SW exchange fluxes. The analysis and conclusion of the paper are based on depth-resolved measurements of saturated sediment thermal conductivities (ke) and the inverse modelling of observed sediemnt temperatures.

The paper is generally well written and presents original data. The authors discuss their findings in the light of the numerous other studies in the field of heat as a natural hydrologic tracer. While there are no ground-braking new results , the paper contributes to further constrain the uncertainties associated with thermal conductivity estimation in heat tracing studies.

Specific comments:

[Figure]

p.3. l.12-14. This sentence is redundant to the one in p.2. l. 31. , consider to remove/rephrase Section 4.1. The reported thermal conductivities of partially <0.6 W/m/K are lower than those of pure water. Could this be attributed to accidently unsatured conditions? Otherwise such low values seem very unlikely if not physically impossible in saturated sediments. The low values should be discussed in Section 5.2.

Section 4.2. and Fig. 3. The measured temperature-depth profiles, including the cases with poor model fits, seem to reasonably represent a steady state case with upward water flow. I wonder if the depth of the domain (only 1m) and the selected lower temperature boundaries are really appropriate. My impression is that the boudary conditions are too rigid to provide a good fit. For example: in Fig. 3 - P1 the lower temperature boundary seems too low. Maybe extend the model domain to greater depths or use the lowest temperature measurements as boundary condition.

p.7.l.18 and following. ke and vertical water fluxes(qz) are related. In steady-state 1D, homogeneous conditions there should be functional relationship between qz and ke. I suggest to present the results along the theoretical relationship. Then it would also be possible to evealuate/visualize the effect of heterogeneous vs homogeneous ke

p.8. l.21-28. Maybe the limited spatial resolution of the measurements calls for a geostatistical approach, similarly to generation hydraulic conductivity fields, to come up with spatially continuous scenarios of ke. Maybe briefly discuss this option.

p.9. l. 21. Does ke really increase with grain size? If porosity and the sediment material do not change one would expect ke to be constant (if one assumes that ke of the water-sediment mixture can be modelled by the volume fractions and the thermal conductivities of water and sediment grains). An alternative explanation for the observation could be that the shallow sediments are less consolidated and have a higher porosity which could explain the lower thermal conductivity. I think, as porosity was not measured, the porosity-dependence should be mentioned and dsicussed.

Technical comments:

p.5 l.4. better "within" instead of "in"

Figure 1. Add a scale to the insets in b and c

Figure 4. Cases should be "thermal conductivity" not diffusivity

———————————————————

---

## Referee Comment (RC2) · Anonymous Referee #2 · 18 Jun 2018

Author comments

General comments: The manuscript "The effect of sediment thermal conductivity on vertical groundwater flux estimates" used measured profiles of sediment temperatures and bulk thermal conductivities (ke, using a KD2Pro thermal property analyser) with depth in two contrasting environments, and used these data in conjunction with Hydro-GeoSphere (HGS) and PEST to determine upwelling fluxes. The analyses investigated the use of the detailed ke profiles as well as homogeneous profiles on the resulting fluxes from HGS.

Overall, the manuscript was interesting to read, well written and clearly explained. The figures were also of a high quality.

Specific comments: The temperature-depth profiles are taken at a specific point in time. Presumably the profiles at a particular site were all taken within a short time frame? At any rate, the use of steady state temperatures is likely an additional source of uncertainty in these analyses. There is an equation presented in Briggs et al. (2014, JoH) that can be used to determine the propagation depth of a diurnal signal. This could be used to determine whether transience is likely to be influencing the temperature profile at each depth. Presumably the upper part of all profiles is not in steady state, especially the lower flux site. An investigation into the implications of this, and comments on the influence of transience in the temperature profiles would be useful.

There are a number of numerical modelling programs that are custom made to fit temperature data to determine fluxes (e.g. Munz and Schmidt, 2017 HP, Koch et al. 2015, GW). Is there any particular reason why HGS was used over these other approaches?

I think that the selected boundary conditions in the HGS simulations are also a major source of uncertainty/error. Rather than setting the water temperature at z = 0 and a deeper groundwater temperature, why not use the measured temperatures at the top and bottom of the profile as the boundary conditions? This would dramatically improve the fits on some of these profiles (e.g. P4, upper part of S4, P1, S7, H4). This will likely significantly change the resulting flux estimates. The large mismatch between observed and modelled data look to be a major source of uncertainty.

It would also be useful to see the T-z profiles from all (or more) of the sites. In particular, the low flux environments. Alternatively, a way to show the RMSE that goes with the values in Fig3 and Fig4 would help show whether poor fits are a major source of error or not.

Page 2 lines 6-7, there are also time series based methods for mapping fluxes (e.g. Lautz and Ribaudo 2012, HJ, Irvine and Lautz 2015 JoH).

Page 2, lines 24-25: The McCallum/Luce methods do not require thermal conductivity to estimate fluxes. They can also be used to determine thermal conductivity. i.e. these

are two separate approaches. It is not immediately clear if this is what is meant in the first two sentences here.

Technical corrections: Page 9, lines 23-25: In the sentence about the paper from Duque et al, is this depth supposed to be 0 m?

---

## Author Comment (AC1) · 9 Jul 2019

**Response to reviewers' comments to the manuscript:" The effect of sediment thermal conductivity on vertical groundwater flux estimates, MS number: hess-2018-210**

**First of all the authors would like to thank the two anonymous reviewers for the encouraging and useful comments! Based on the suggestions we believe that we managed to address all concerns of the reviewers and generally improve the clarity of the manuscript.**

**Please note that the references to page, line and figure numbers in the corrected manuscript refer to the revised manuscript submitted together with this response.**

**Response to Referee #1:**

General comments: The paper presents an evaluation of the influence of vertical thermal conductivity variability on the estimates of vertical GW-SW exchange fluxes. The analysis and conclusion of the paper are based on depth-resolved measurements of saturated sediment thermal conductivities (ke) and the inverse modelling of observed sediment temperatures.
The paper is generally well written and presents original data. The authors discuss their findings in the light of the numerous other studies in the field of heat as a natural hydrologic tracer. While there are no ground-braking new results, the paper contributes to further constrain the uncertainties associated with thermal conductivity estimation in heat tracing studies.

Specific comments:
p.3. l.12-14. This sentence is redundant to the one in p.2. l. 31.

**Action: Sentence at p.3, l. 12-14 removed.**

Consider to remove/rephrase Section 4.1. The reported thermal conductivities of partially <0.6 W/m/K are lower than those of pure water. Could this be attributed to accidently unsaturated conditions? Otherwise such low values seem very unlikely if not physically impossible in saturated sediments. The low values should be discussed in Section 5.2.

**The thermal conductivity of sediments is influenced by the density, moisture content of the sediments, also the salinity of pore water and the content of organic matter in the sediment material (Abu-Hamdeh and Reeder, 2000). During the field measurements some of the sediment cores became unsaturated (p.5., l. 4-5) and sediment thermal conductivity values were therefore removed from the analysis.**
**Both at the lagoon and at the stream site organic matter and plant debris was also occasionally trapped in the sediment columns, close to the sediment surface at shallow depths. Thus it is assumed that in some cases organic matter decreased sediment thermal conductivity. Pooling all thermal conductivity values together, four measurements gave a thermal conductivity below 0.73 W m$^{-1}$ °C$^{-1}$ and three of these measurements were made at the stream site which is known to have organic debris also deeper in the sediment column (Sebok et al., 2014). As neither unsaturated conditions, nor organic sediments were visually identified for these samples and the measurement error was within the chosen limits of the**

study (0.05 W m$^{-1}$ $^{\circ}$C$^{-1}$), the authors did not find any rigorous reason to remove these values from the validated measurements.

**Action: As Section 4.1 only presents our validated results we chose not to change the text and discuss the issue in Section 5.2.**
**Text in Section 5.2 was rephrased, now including:** '*At the stream site unusually low sediment thermal conductivity values between 0.55 and 0.65 W m$^{-1}$ $^{\bullet}$C$^{-1}$ were observed. These values are clearly outliers in their respective measurement depths (Fig. 2). However, as the sediment core did not become unsaturated, nor the measurement error was too high to discard the measurement, it is assumed that sediment organic matter resulted in such a low thermal conductivity value which was previously shown to be occasionally present also deeper in the stream sediments (Sebok et al.,2014).*' **(p.9., l. 27-31.)**

Section 4.2. and Fig. 3. The measured temperature-depth profiles, including the cases with poor model fits, seem to reasonably represent a steady state case with upward water flow. I wonder if the depth of the domain (only 1m) and the selected lower temperature boundaries are really appropriate. My impression is that the boundary conditions are too rigid to provide a good fit. For example: in Fig. 3 - P1 the lower temperature boundary seems too low. Maybe extend the model domain to greater depths or use the lowest temperature measurements as boundary condition.

**In answering this comment we would like to refer to each field site separately. At the stream site, at the high discharge zone the upward groundwater flux is high enough for reaching stable groundwater temperatures at 1 m depth below the streambed surface as also presented by field measurements in other studies (Karan et al., 2013; Jensen and Engesgaard, 2011), thus in case of the stream site we do not think it is necessary to change the depth of the lower temperature boundary condition. Especially as the RMSE of the temperature profiles is between 0.02 and 0.32 °C, while the measurement accuracy was 0.2 °C.**

**At the lagoon site upward groundwater fluxes are lower, thus stable groundwater temperatures will not be reached at 1 m depth below the lagoon surface where we set the lower temperature boundary. We have however several reasons to maintain the temperature boundary condition at 1 m depth below the lagoon surface:**
- **As already discussed in the manuscript text (p. 8, l. 28 – p. 9, l. 1), in the low flux lagoon site assuming only vertical flow conditions may not be correct as wave action can also induce a temporary horizontal flow component in shallow depths. Moreover, the diurnal variations in air temperature are more pronounced in the upper part of the temperature profiles (for a more precise description please refer to the response given to Referee #2). If we use the measured temperatures at 0.5 m depth as a boundary condition, we can only fit the model to temperature data collected up to 0.35 m depth, which is shallow enough to be exposed both to a horizontal flow component and diurnal temperature variations. For this reason we would argue against moving the model boundaries up to the temperatures measured at 0.5 m depth.**
- **In the lagoon at greater depths density-driven flow also induces a strong horizontal groundwater flow component by the movement of the saline wedge that varies depending on the season and recharge conditions. Based on field data, Müller et al. (2018) estimated the depth of the density driven flow at approx. 2 m below the lagoon surface, thus moving the model boundary deeper than 1 m would also introduce additional uncertainty to the flux estimates.**

- **Sediment temperature was measured at 7 locations (0, 5, 10, 15, 20, 35, 50 cm depth) below the lagoon surface. Using the temperatures measured at 0 cm and 50 cm depth as boundary conditions would also mean that we only can evaluate the fit between observed and simulated data at 5 depths, where four of the measurement points are only 20 cm below the lagoon surface. As this area is the most affected by the diurnal temperature changes, we think that we also need the temperature data at 50 cm depth to have a more robust flux estimate and also to include as much of the measured data in the estimation process as possible.**
- **Selecting the temperature boundary condition at 1 m below the lagoon bed is also a good way to minimize boundary effects, while using temperature data at 0.5 m depth would introduce an even more rigorous boundary condition, thus influence flux estimates in a higher degree. As an example at profile P1 using the temperatures measured at 0.5 m depth below the surface as a lower boundary condition would increase the obtained flux values in such a degree that they are not realistic anymore. For profile P1, this would result in an increase from 0.17 m/d to 0.35 m/d. Having several years of field work experience at the site (Haider et al., 2014; Duque et al., 2016) the authors carried out numerous temperature profile-based and seepage meter based flux estimates which never showed such high flux values at the lagoon.**
- **Our most important argument about using the presented boundary condition is that our aim with the manuscript was to conceptualize the effect of using various, even vertically heterogeneous distributions of measured sediment thermal conductivity and study their effect on flux estimates. Using the same temperature boundary conditions at the same depth provides a common background to all measured temperature profiles at the respective field sites. We feel that using different temperature boundary conditions for profiles measured 10-15 minutes and 1 m apart would not provide for a stable background for comparison. Furthermore, our interest lies in the differences between flux estimates within individual profiles using different sediment thermal conductivities, instead of describing the spatial variability of flux estimates within different temperature profiles. For the within-profile comparison, results are representative if the same boundary conditions are used for all cases of different sediment thermal conductivities. Thus, we think that irrespective of the RMSE of the profiles, the change in the RMSE while using different sediment thermal conductivities is sufficient to make conclusions about the effect of using different sediment thermal conductivities on vertical flux estimates.**

**In order to test the effect of the depth and temperature of the boundary condition on the flux estimates, we reanalyzed profile P1 from the lagoon which had the one of the worst RMSE values of all profiles in this study assuming the average sediment thermal conductivity measured in the profile.**

- **Using the sediment temperature measured at 0.5 m depth resulted in a flux estimate of 0.35 m/d with an RMSE of 0.37 °C. Thus the authors would argue against using the measured sediment temperature at 0.5 m depth as a lower boundary condition due to the unreasonably large flux estimate**
- **Using a common, assumed groundwater temperature of 11.5 °C at different depths, the following flux estimates and RMSE were obtained with an analytical solution:**

| Depth of stable groundwater temperature (m) | Flux (m/d) | RMSE (°C) |
|---|---|---|
| 0.5 | 0.15 | 1.00 |
| 1 | 0.16 | 0.77 |
| 1.5 | 0.16 | 0.75 |
| 2 | 0.16 | 0.75 |
| 3 | 0.16 | 0.75 |
| 4 | 0.16 | 0.75 |
| 5 | 0.16 | 0.75 |

**Thus assuming a constant groundwater temperature at greater depth than 1 m would not considerably improve the RMSE of the profile, while the flux values stay constant. Raising the constant temperature boundary to 0.5 m would on the other hand increase RMSE and result in unreasonably high fluxes.**

**Based on both the theoretical considerations and the results obtained in profile P1 we would argue against changing the depth of the boundary condition as in a greater depth the RMSE improves slightly, but more uncertainty is introduced in the profiles by entering the zone of the density-driven flow dynamics.**

**No action**

p.7.l.18 and following. ke and vertical water fluxes(qz) are related. In steady-state 1D, homogeneous conditions there should be functional relationship between qz and ke. I suggest to present the results along the theoretical relationship. Then it would also be possible to evevaluate/visualize the effect of heterogeneous vs homogeneous ke.

**There is certainly a functional relationship between $k_e$ and $q_z$ (Figure 1, in response) which is clearly visible assuming a homogeneous distribution of $k_e$ through the vertical sediment column. Our intention in the manuscript however was to present the different flux values that can be obtained by using actual $k_e$ measurements within one single profile within real field settings rather than a theoretical range of potential $_{ke}$ values. This way the emphasis of the study is not on how much the fluxes change when assuming a range of $_{ke}$ values, but the fact that such a large range of $_{ke}$ values could be measured within the profiles thus highlighting the importance of selecting an appropriate $_{ke}$ value for flux calculations.**

**No action**

[Figure]

**Figure 1: The functional relationship between ke and q derived from the Peclet number.**

p.8. l.21-28. Maybe the limited spatial resolution of the measurements calls for a geostatistical approach, similarly to generation hydraulic conductivity fields, to come up with spatially continuous scenarios of ke. Maybe briefly discuss this option.

**This is an interesting point made by the Referee. In the text (p.8 1. 24-25) we highlight that the vertical natural variability in the sediments may be higher than what we sample. We have several reasons, why we did not include geostatisctial approaches creating e.g. variograms in the manuscript:**
   **i) From a geostatistical point of view only an appropriate sample size can create meaningful variograms. Eventhough our data is of relatively high resolution compared to previous studies, there are still too few datapoints in vertical direction to generate meaningful vertical variograms.**
   **ii) To overcome such a problem we could bin all observations together. But that would require similar sedimentation conditions and spatially continuous data. Both of these requirements are violated by the three different measurement sites as well as the different depositional environments: stream environment, open lagoon, protected lagoon bay.**

**At the same time we attempted a geostatistical approach in case of the peat profiles of the lagoon.**
   **i) From the test variogram, the calculated range was very short (Figure 2, in response), on the scale of 0.2 m.**
   **Hence, we would argue that geostatistical approaches similar to hydraulic K field generation would be largely biased by the few vertical datapoints collected**

**Moreover its application to the present environment may be inappropriate. As this natural environment is characterized by large heterogeneity occurring due to small-scale faunal activity (worm or crab activity etc.), rooting of plants disturbing sediment structures or erosional events caused by storm wave activity rearranging the natural settling conditions expected in near coastal zones. Furthermore, all those factors influence the natural setup on a very short temporal scale (especially tidal and wave actions).**

**No action**

[Figure]

[Figure]

**Figure 2 Geostatistical exploration of ke at the lagoon sites. Upper panel shows the variogram of the log.values of ke. A very short range of 0.2m is established and thereby a low vertical spatial relation is achieved. After a distance of 0.2 m no spatial relation can be established between the values. The lower panel shows krieged horizontal ke surfaces at different depths using the exponential model. Here a large variability of values in each separated depth can be seen. However, due to the few datapoints per depth the resulting spatial statistics may be highly biased.**

p.9. l. 21. Does ke really increase with grain size? If porosity and the sediment material do not change one would expect ke to be constant (if one assumes that ke of the water-sediment mixture can be modelled by the volume fractions and the thermal conductivities of water and sediment grains). An alternative explanation for the observation could be that the shallow sediments are less consolidated and have a higher porosity which could explain the lower thermal conductivity. I think, as porosity was not measured, the porosity-dependence should be mentioned and dsicussed.

**We agree with the Referee that sediment thermal conductivity k$_e$ depends on porosity, which is related to grain size and packing conditions.**

**Action: The manuscript text was rephrased to: "*An explanation for this could be that measurements in this study were also made at other depths below the SWI, where thermal conductivity values show a generally increasing trend with depth. This is likely to reflect a transition from finer, less consolidated sediments of higher porosity to coarser, more consolidated sediments of lower porosity.*" Page 9 line 20-23**

Technical comments:
p.5 l.4. better "within" instead of "in"

**Action: Changed**

Figure 1. Add a scale to the insets in b and c

**Scale added to the insets.**

Figure 4. Cases should be "thermal conductivity" not diffusivity

**Figure inscription corrected.**

References:

Abu-Hamdeh, N. H. and Reeder, N. C.: Soil Thermal Conductivity: Effects of Density, Moisture, Salt Concentration, and Organic Matter, Soil Sci. Soc. Am. J. 64:1285–1290, 2000.

Duque, C., Müller, S., Sebok, E., Haider, K. and Engesgaard, P.: Estimating groundwater discharge to surface waters using heat as a tracer in low flux environments: The role of thermal conductivity, Hydrol. Proc., 30(3), 383–395, 2016.

Haider, K., Engesgaard, P., Sonnenborg, T. O. and Kirkegaard, C.: Numerical modeling of salinity distribution and submarine groundwater discharge to a coastal lagoon based on airborne electromagnetic data, Hydrogeology Journal, DOI:10.1007/s10040-014-1195-0, 2014.

Jensen, J. K. and Engesgaard, P.: Nonuniform groundwater discharge across a Streambed: Heat as a tracer, Vadose Zone J., 20 10, 98–109, doi:10.2136/vzj2010.0005, 2011.

Karan, S., Engesgaard, P., Looms, M. C., Laier, T. and Kazmierczak, J.: Groundwater flow and mixing in a wetland-stream system: Field study and numerical modelling, J. Hydrol., 488, 73-83, 2013.

Müller, S., Engesgaard, P., Jessen, S., Duque, C., Sebok, E. and Neilson, B.: Assessing seasonal flow dynamics at a lagoon saltwater-freshwater interface using a dual tracer approach, Journal of Hydrology, Regional Studies, 17, 24-35, 2018.

Sebok, E., Duque, C., Engesgaard, P. and Boegh, E.: Spatial variability in streambed hydraulic conductivity of contrasting stream morphologies: channel bend and straight channel, Hydrol. Process., 29 (3), 458-472, doi:10.1002/hyp.10170, 2014.

---

## Author Comment (AC2) · 9 Jul 2019

**Response to reviewers' comments to the manuscript:" The effect of sediment thermal conductivity on vertical groundwater flux estimates, MS number: hess-2018-210**

**First of all the authors would like to thank the two anonymous reviewers for the encouraging and useful comments! Based on the suggestions we believe that we managed to address all concerns of the reviewers and generally improve the clarity of the manuscript.**

**Please note that the references to page, line and figure numbers in the corrected manuscript refer to the revised manuscript submitted together with this response.**

**Response to Referee #2:**

General comments: The manuscript "The effect of sediment thermal conductivity on vertical groundwater flux estimates" used measured profiles of sediment temperatures and bulk thermal conductivities (ke, using a KD2Pro thermal property analyser) with depth in two contrasting environments, and used these data in conjunction with Hydro-GeoSphere (HGS) and PEST to determine upwelling fluxes. The analyses investigated the use of the detailed ke profiles as well as homogeneous profiles on the resulting fluxes from HGS.
Overall, the manuscript was interesting to read, well written and clearly explained. The figures were also of a high quality.

Specific comments:
The temperature-depth profiles are taken at a specific point in time. Presumably the profiles at a particular site were all taken within a short time frame? At any rate, the use of steady state temperatures is likely an additional source of uncertainty in these analyses. There is an equation presented in Briggs et al. (2014, JoH) that can be used to determine the propagation depth of a diurnal signal. This could be used to determine whether transience is likely to be influencing the temperature profile at each depth. Presumably the upper part of all profiles is not in steady state, especially the lower flux site. An investigation into the implications of this, and comments on the influence of transience in the temperature profiles would be useful.

**The temperature profiles were taken within a time interval of a few hours at each measurement site, thus transience in the upper part of the profiles can be expected. At the stream site however, as the majority of stream water is originating from groundwater (thus having a relatively stable temperature) and due to the high velocity water flow, the high upward groundwater fluxes and the thickness of the water column, the transience in the upper part of the sediment profiles is negligible.**
**In the low-flux, shallow lagoon environment however, transience can be more pronounced. The effect of transience was therefore assessed at the lagoon site using the analytical solution (Goto et al. 2005) reported in Briggs et al. (2014) under the current field settings (see table below), assuming only heat conduction. The results show that the propagation depth of the diurnal signal will be measurable only until a depth of 0.1 m below the sediment bed when assuming extreme boundaries of 5 degree temperature amplitude and a 1h response time (Figure 1, in response). However, such assumptions are unlikely to occur in natural settings. Under natural field conditions upward fluxes can be expected to shift the propagation depth**

higher up towards the sediment-water interface. Additionally, lowering the thermal conductivity will minimize the propagation depth and vice versa. Such low thermal conductivities were typically observed in the shallowest parts of the profiles (Fig.2 in the manuscript).

Thus in the timeframe the measurements were taken, the upper part of the sediment temperature profiles can be assumed to be in steady state.

|  |  | unit |
|---|---|---|
| thermal conductivity: | 1.8 | [J/m s °C] |
| fluid heat capacity: | 4192 | [J/kg °C] |
| fluid density: | 999.73 | [kg/m$^3$] |

**Table 1: Input parameters for the Stallman model**

[Figure]

Figure 1: Propagation of the diurnal temperature signal in the lagoon bed, assuming the measured thermal parameters(Table 1, in response) at the lagoon and a temperature amplitude of up to 5 ºC (left) and a time interval between 1 and 24 hours (right).

**Action: Results of test calculating the penetration depth was added to the manuscript text:**
'*Using the solution presented by Briggs et al. (2004) with the thermal parameters measured in the lagoon assuming 5º C diurnal amplitude and only heat conduction, the penetration depth of the diurnal signal was found to be 0.1 m under the lagoon bed. Due to the upward fluxes at the lagoon this penetration depth is even shallower, thus it is assumed that transience in the temperature profiles does not affect results significantly.*' **Page 9 lines 1-4**

There are a number of numerical modelling programs that are custom made to fit temperature data to determine fluxes (e.g. Munz and Schmidt, 2017 HP, Koch et al. 2015, GW). Is there any particular reason why HGS was used over these other approaches?

**HydroGeoSphere was selected as a modelling program as a similar code coupled with PEST was already available to the authors from a previous study.**

**No action**

I think that the selected boundary conditions in the HGS simulations are also a major source of uncertainty/error. Rather than setting the water temperature at $z = 0$ and a deeper groundwater temperature, why not use the measured temperatures at the top and bottom of the profile as the boundary conditions? This would dramatically improve the fits on some of these profiles (e.g. P4, upper part of S4, P1, S7, H4). This will likely significantly change the resulting flux estimates. The large mismatch between observed and modelled data look to be a major source of uncertainty.

**The reviewer is referred to the response given to the comment of Referee# 1 on Section 4.2 and Figure 3.**

It would also be useful to see the T-z profiles from all (or more) of the sites. In particular, the low flux environments. Alternatively, a way to show the RMSE that goes with the values in Fig3 and Fig4 would help show whether poor fits are a major source of error or not.

**Our intention with including Figure 3 in the manuscript was to visualize the T-z profiles and provide an opportunity to the readers to assess the fit between the measured and simulated data. For this reason for each measurement site we selected the profile with best and worst fit between observed and simulated data and also included in the manuscript text the best and worst RMSE values for the five cases (page 6, line 27-31). As each measurement profile would have 6 datasets on the T-z figure (measured data and the five cases) we believe that a separate figure would be needed for each individual profile in order to maintain the readability of the figure. Furthermore as the included profiles are typical for the measurement sites we feel that providing an extra figure would not give any additional value to our manuscript.**

**No action**

Page 2 lines 6-7, there are also time series based methods for mapping fluxes (e.g. Lautz and Ribaudo 2012, HJ, Irvine and Lautz 2015 JoH).

**Action: Reference to the study of Lautz and Ribaudo (2012) added to the manuscript: "*The temperature distribution at the bed of surface water bodies can be used for qualitative mapping of potential discharge sites (Conant, 2004; Sebok et al., 2013; Briggs et al., 2011) or supplemented by heat transport modelling also for obtaining flux estimates over larger areas (Lautz and Ribaudo, 2012)." Page 2 line 7-10**

Page 2, lines 24-25: The McCallum/Luce methods do not require thermal conductivity to estimate fluxes. They can also be used to determine thermal conductivity. i.e. these are two separate approaches. It is not immediately clear if this is what is meant in the first two sentences here.

**Action: The manuscript text was changed to clarify this misunderstanding: '*For some approaches sediment thermal conductivity (ke) is not required to estimate groundwater flux and in a separate approach sediment temperature time series can be used to estimate sediment thermal diffusivity (McCallum et al., 2012; Luce et al., 2013).' Page 2 line 25-27**

Technical corrections: Page 9, lines 23-25: In the sentence about the paper from Duque et al, is this depth supposed to be 0 m?

**Action: Sentence was rephrased to: "***Previously, Duque at al. (2016) also measured thermal conductivities between 0.62-2.19 W/m•C at the surface of the lagoon bed at 0 m depth, while in our study values between 0.65 and 1.99 W/m•C were found at 0 m depth at the lagoon surface.***"** **Page 9 line 25-27**

References:
Briggs, M. A., Lautz, L. K., Buckley, S. F., and Lane J. W.: Practical limitations on the use of diurnal temperature signals to quantify groundwater upwelling, J. Hydrol., 519, 2014.
Goto, S., Yamano, M. and Kinoshita M.: Thermal response of sediment with vertical fluid flow to periodic temperature variation at the surface, J. Geophys. Res., 110, 2005.